# Bone Involvement in Rheumatoid Arthritis and Spondyloartritis: An Updated Review

**DOI:** 10.3390/biology12101320

**Published:** 2023-10-09

**Authors:** Francesco Orsini, Chiara Crotti, Gilberto Cincinelli, Raffaele Di Taranto, Andrea Amati, Matteo Ferrito, Massimo Varenna, Roberto Caporali

**Affiliations:** 1Department of Clinical Sciences and Community Health, Università degli Studi di Milano, 20122 Milan, Italyandrea.amati@unimi.it (A.A.);; 2Department of Rheumatology and Medical Sciences, ASST G.Pini-CTO, 20122 Milan, Italy; 3Bone Diseases Unit, Department of Rheumatology and Medical Sciences, ASST G.Pini-CTO, 20122 Milan, Italy

**Keywords:** osteoimmunology, rheumatoid arthritis, erosions, spondyloarthritis, bone tissue, new bone formation

## Abstract

**Simple Summary:**

Chronic inflammatory arthritis, such as rheumatoid arthritis (RA) and spondyloarthritis (SpA), often have a significant impact on bone tissue, where bone is not just a passive target but actively contributes to the disease progression. This review explores the pathogenic mechanisms involving bone, highlighting the complex molecular interactions between bone cells and the immune system, a field known as osteoimmunology. It discusses the unique processes of bone erosion and systemic bone loss in RA and SpA, as well as abnormal bone formation in SpA.

**Abstract:**

Several rheumatologic diseases are primarily distinguished by their involvement of bone tissue, which not only serves as a mere target of the condition but often plays a pivotal role in its pathogenesis. This scenario is particularly prominent in chronic inflammatory arthritis such as rheumatoid arthritis (RA) and spondyloarthritis (SpA). Given the immunological and systemic nature of these diseases, in this review, we report an overview of the pathogenic mechanisms underlying specific bone involvement, focusing on the complex interactions that occur between bone tissue’s own cells and the molecular and cellular actors of the immune system, a recent and fascinating field of interest defined as osteoimmunology. Specifically, we comprehensively elaborate on the distinct pathogenic mechanisms of bone erosion seen in both rheumatoid arthritis and spondyloarthritis, as well as the characteristic process of aberrant bone formation observed in spondyloarthritis. Lastly, chronic inflammatory arthritis leads to systemic bone involvement, resulting in systemic bone loss and consequent osteoporosis, along with increased skeletal fragility.

## 1. Introduction

Bone tissue is frequently involved in rheumatologic diseases, particularly chronic inflammatory diseases, such as rheumatoid arthritis (RA) and spondyloarthritis (SpA). In recent years, osteoimmunology has provided extensive evidence for the role of skeletal tissue in the pathogenesis of many rheumatological diseases.

The intricate changes in bone tissue in these conditions result in diverse bone phenotypes. In RA, bone tissue is characterized by an uncoupling between the bone formation and bone resorption processes, resulting in a significant increase in osteoclastic activity, leading to an erosive bone phenotype that clinically manifests as skeletal erosions. Conversely spondyloarthritis is characterized by a bone phenotype that encompasses both erosive and osteoproliferative aspects, giving this group of diseases peculiar and only partially known characteristics.

In addition to local structural damage, chronic arthritis is also characterized by systemic involvement resulting in skeletal fragility secondary to systemic osteoporosis.

The aim of this review is to provide a comprehensive overview of the pathogenetic mechanisms underlying bone tissue involvement in chronic arthritis.

## 2. Bone Remodeling and Homeostasis in Health

Under physiological conditions, bone tissue structure and mass are maintained through a timely balanced process called bone remodeling. Bone remodeling consists of the tightly coupled actions of bone removal and apposition conducted by resident specialized cells: osteoclasts and osteoblasts, respectively [1].

Bone-remodeling cellular events take place in a chronologically hierarchical fashion, with osteoclasts demineralizing and eroding the bone matrix in discrete anatomical areas called resorption lacunae (or Howship lacunae), followed by the apposition of new matrix components and subsequent matrix mineralization completed by osteoblasts [2].

Osteoclasts are fused, multinucleated cells of hematopoietic origin. Two molecules are essential for osteoclast differentiation. The first step of osteoclast precursors’ proliferation and maturation towards the monocytes/macrophages lineage is driven by the macrophage-colony stimulating factor (M-CSF) [3]. Subsequent osteoclastogenesis processes are determined by the binding of the tumor necrosis factor (TNF)-superfamily cytokine receptor activator of the nuclear factor kappa B (NF-κB) ligand (RANKL) to the receptor activator of NF-κB (RANK) expressed on pre-osteoclasts membranes. RANK-RANKL interaction allows for downstream signaling pathways of NF-κB, activator protein 1 (AP-1), and Nuclear Factor of Activated T Cells 1 (NFATc1), and ultimately leads to cellular fusion and final osteoclast maturation and activation [4,5]. RANKL is mainly produced by stromal cells such as osteoblasts, osteocytes, and fibroblasts, as well as by immune cells [5,6]. To downwardly tune the osteoclast activation and bone removal in healthy conditions, a decoy receptor called osteoprotegerin (OPG) is produced by osteoblasts to disrupt the RANK-RANKL interaction [7].

Bone-forming cells’ osteoblasts originate from pluripotent mesenchymal stem cells (MSC), guided in their commitment by different signals such as the parathyroid hormone (PTH), vitamin D 1,25(OH), transforming growth factor (TGF)-β, and fibroblast growth factor (FGF), as well as mechanical loading [8,9,10,11]. These factors promote the cascade of the bone morphogenic protein (BMP) and the Wingless and int-1 (Wnt)/β-catenin pathways, vital intracellular mechanisms that positively regulate the expression of runt-related factor 2 (Runx2) and osterix to drive osteoblasts maturation [12] and molecules like OPG and alkaline phosphatase (ALP) to exert their bone-forming properties [13]. Appropriate regulation of osteoblasts’ differentiation relies upon three main inhibitor proteins. Sclerostin, mainly produced by osteocytes, and the proteins member of the Dickoppf (Dkk) family act as antagonists of the low-density lipoproteins (LPR)-5/6, essential Wnt co-receptors [14], while the secreted frizzled-related protein family (sFRP) and Wnt inhibitory factor (WIF-1) inhibit the interaction between Wnt and its receptor [15].

A key director role in bone remodeling is played by osteocytes, former osteoblasts embedded throughout the bone matrix during the process of bone deposition. As they become entombed, osteocytes’ cellular bodies situated deep into the lacunae change shape and grow dendritic processes that run capillary-like into canaliculi to reach periosteal and endosteal surfaces and other lacunae, to communicate with other bone cells (osteoblasts, osteoclasts, and osteocytes), blood vessels, and with the bone marrow stroma [16]. In this way, osteocytes can sense both mechanical stimuli and local and circulating factors to regulate the bone remodeling processes through the calibrated release of sclerostin, OPG, and RANKL [17].

Lastly, the bone cells and microenvironment are enriched by connections with and regulation by immune cells and their products, above all cytokines [18]. The bone-immunity interplay is capable of tuning the abovementioned bone resorption and formation mechanisms, laying the pathogenetic ground for the bone disease observed in rheumatic disorders such as inflammatory arthritis [19].

## 3. Bone Involvement in Rheumatoid Arthritis

### 3.1. Immune Cells

Contrary to other resident cells in the bone, osteoclasts stem from progenitors of the monocytic/macrophage lineage, specifically from pre-osteoclastic progenitors within the bone or from circulating peripheral precursors such as monocytes. However, several in vitro studies in the presence of an inflammatory milieu [20,21] and in vivo data from animal models of aseptic bone inflammation [22] have demonstrated the ability of dendritic cells to transdifferentiate into cells with an osteoclast-like phenotype, fueling bone remodeling imbalance and subsequent net bone loss. Dendritic cells (DC) have the same embryological origin and share certain cellular markers with osteoclasts, such as αvβ3 integrin, an integral membrane glycoprotein that exerts cell fusion for DCs and adhesion to the bone surface at the site of resorption for osteoclasts. It is not yet known whether the differentiation of dendritic cells into osteoclast-like cells is solely mediated by the Receptor Activator of NF-κB Ligand (RANK-L) and Macrophage Colony-Stimulating Factor (M-CSF), or if there are additional factors specific to the inflammatory microenvironment that actively contribute to this process [23]

In recent years, the newly emerging field of osteoimmunology has received increasing attention regarding the multiple and tangled interactions between the immune and skeletal systems in physiologic and pathological conditions. One of the key aspects of osteoimmunology is the interaction between T cells and bone homeostasis (Table 1), for example through the RANK/RANK-L/OPG pathway. T cells can differentiate into different T helper (Th) cell subsets: Th1, Th2, Th17, and regulatory T cells (Tregs), each with distinct effects on bone metabolism. It was observed that the Th1 and Th2 subsets inhibit the RANK-L- and M-CSF-dependent differentiation of osteoclast progenitors [24] in an in vitro model of osteoclastogenesis. In the context of osteoclastogenesis, Interferon- γ (IFN-γ) and interleukin-4 (IL-4), mainly produced by Th1 and Th2 cells, respectively, can inhibit the expression of the RANK receptor on osteoclast progenitors and reduce their sensitivity to the differentiation signals induced by RANK-L. Conversely, Th17 cells have been shown to promote osteoclastic activity through the secretion of IL-17. This cytokine does not directly act on osteoclast precursors, but induces the expression of RANK-L by osteoblasts, thereby promoting osteoclast differentiation and activity. Treg cells play a central role in maintaining homeostasis and immunological tolerance and exert a protective effect on bone tissue regardless of the presence of an inflammatory microenvironment [25], prompting a direct effect on the bone remodeling process. In patients with RA, Treg lymphocytes are functionally altered. However, it is not yet fully understood how this may impact bone loss [26].

### 3.2. Cytokines

The pro-inflammatory cytokine orchestra produced by macrophages, T lymphocytes, and synovial fibroblasts plays a crucial role in the bone involvement in RA, both in the genesis of focal bone damage and in systemic bone loss (Figure 1). The central inflammatory cytokines involved are Tumor Necrosis Factor-α (TNF-α), IL-1β, and IL-6, targeted by biotechnological drugs that have dramatically improved the prognosis of patients with RA unresponsive to conventional disease-modifying antirheumatic drugs (DMARDs) therapy.

#### 3.2.1. TNF-α

TNF-α significantly alters bone remodeling through different mechanisms. This cytokine acts indirectly on the process of osteoclastogenesis by promoting the expression of RANKL, a member of the TNF superfamily, and by stimulating the autocrine/paracrine expression of IL-1 and Interleukin-1 Receptor 1 (IL-1R1) [29]. A significant portion of the osteoclast-inducing activity of TNF-α is mediated by IL-1, as evidenced by the approximately 50% reduction in TNF-induced osteoclastogenesis observed in IL-1R1-deficient mice [30]. This result highlights the reinforcing and synergistic interplay between these two inflammatory cytokines in promoting bone resorption. IL-1 is the pro-inflammatory cytokine with the highest dose-dependent osteoclastogenic effect in monocellular cultures of mouse bone marrow-derived macrophages at permissive concentrations of RANKL, compared with TNF-α, IL-6, IL-17, and IL-23 [29]. Furthermore, TNF-α might exert a direct, RANKL-independent, effect in promoting the differentiation of mouse monocytic-macrophage lineage precursors towards an osteoclast phenotype. This effect was observed when TNF-α was added to the culture system of osteoclastogenesis at lower concentrations of RANKL [29].

When the direct effect of TNF-α on an osteocyte population was investigated, it was demonstrated that TNF-α induces an increase in the expression of RANKL-positive osteocytes. This effect was associated with the up-regulation of phosphorylation of extracellular signal-regulated Kinases (ERK) 1/2, p38 mitogen-activated protein kinases (p38), and c-Jun N-terminal kinases (JNKs) signaling pathway proteins [31].

The overexpression of TNF-α in RA also occurs on the osteogenic side of bone remodeling with the overexpression of Dickkopf-1 (DKK-1), one of the main negative regulators of the canonical Wnt-β-catenin pathway that is strongly implicated in the processes of proliferation, differentiation, and maturation of osteoblasts. An increased expression of DKK-1 in inflamed synovial tissue in patients with RA was reported [32], speculating on the active role of this mediator in the pathogenesis of bone loss. However, conflicting results have emerged regarding the serum concentrations of this soluble mediator, as a meta-analysis of 136 RA patients and 232 controls found no difference between the two populations [33]. A more recent second meta-analysis [34], which included 9 studies with a total of 1305 RA patients and 504 controls, yielded conclusive evidence of a statistically significant increase in serum concentrations of DKK-1 in patients with RA that contribute to the uncoupling between bone resorption and new bone formation at a systemic level.

The link between TNF-α and sclerostin (SOST), a protein encoded by the SOST gene and produced by osteocytes that impair osteoblastic activity by downregulating the Wnt-β-catenin pathway, is more controversial. Chronic exposure to TNF-α promotes the upregulation of sclerostin and in a rat adjuvant-induced arthritis (AIA) model SOST and DKK-1 were overexpressed in the early stages of arthritis onset [35]. Furthermore, a recent meta-analysis that grouped more than 1000 patients with RA and 561 controls highlighted significantly higher levels of sclerostin in patients with RA [36]. However, both in Sost−/− human TNF-α transgenic (hTNFtg) mice and in hTNFtg mice treated with neutralizing antibodies against murine and human SOST, an exacerbation of arthritic disease has been observed, with an increase in synovial membrane formation and progression in erosive rate [37]. This might suggest a local paracrine action of SOST that acts negatively on TNF-driven inflammation in an inflammatory microenvironment. This issue raised caution regarding the use of romosozumab, a novel monoclonal antibody for the treatment of osteoporosis that binds and inhibits sclerostin, due to a hypothetical worsening of RA and other autoimmune diseases that involve TNF-dependent inflammation. However, neither during the clinical development of the drug nor in subsequent meta-analyses did such a side effect emerge [38,39]. Furthermore, in a study comparing the efficacy of denosumab, a monoclonal antibody targeting RANKL, and romosozumab in patients with RA undergoing chronic treatment with glucocorticoids, a similar improvement in disease activity between the two groups was observed [40].

#### 3.2.2. IL-6

IL-6 is an inflammatory cytokine with a broad range of pleiotropic activities resulting in diverse biological effects on various types of cells. It is a member of a family of cytokines that also includes IL-11, IL-27, IL-31, oncostatin M (OSM), leukaemia inhibitory factor (LIF), ciliary neurotrophic factor (CNTF), cardiotrophin 1 (CT-1), and cardiotrophin-like cytokine factor 1 (CLCF1). IL-6 is a pivotal factor in the inflammation-related joint damage typical of RA; furthermore, this cytokine acts as an acute-phase protein that promotes and intensifies systemic inflammation and its associated comorbidities, such as anemia due to disturbances of iron metabolism or accelerated atherosclerosis. High levels of this inflammatory mediator also affect bone homeostasis favoring the expression of RANKL [41] resulting in systemic bone loss that leads to osteoporosis and an increased risk of fractures [42]. Additionally, the selective overexpression of IL-6 affects bone formation by hindering osteoblast functionality [43]. In an in vitro study IL-6 strongly increased osteoblast apoptosis during the late differentiation stages of murine preosteoblastic cells in culture [44].

#### 3.2.3. OSCAR

Finally, evidence has been reported regarding the action of pro-inflammatory cytokines on the osteoclastic co-stimulation pathway osteoclast-associated receptor (OSCAR)/immunoreceptor tyrosine-based activation motif (ITAM). A secreted form of OSCAR (sOSCAR) has been identified in human blood serum [45]. In this study, the concentration of this soluble mediator was found to be reduced in the serum of patients with RA compared to healthy controls, and it was observed to increase in RA patients treated with TNF inhibitors (TNFi) therapy. This finding has led to scientific speculation on the role of sOSCAR resulting from the cleavage of the extracellular portion of OSCAR, competitively binding to OSCAR ligands and modulating osteoclastogenic stimuli through a negative feedback mechanism. As a result, in chronic inflammatory contexts, TNF-α, IL-1, and IL-6 may inhibit the cleavage of OSCAR and perpetuate the osteoclastogenic signal, thus contributing to inflammatory osteoclastogenesis [46].

### 3.3. Autoantibodies

The role of RA autoimmunity in the pathogenesis of systemic bone loss has been extensively investigated over the years, specifically the action of anti-citrullinated protein (ACPA) antibodies, the Immunoglobulin G (IgG) subclass of antibodies, which target citrullinated proteins such as vimentin, fibrinogen, and collagen. Citrullination is a post-translational modification by which the amino acid arginine is converted into citrulline through the action of specific enzymes called peptidyl arginine deiminases (PAD). ACPA are highly specific for RA, whereas Rheumatoid Factor (RF) can also be found among healthy individuals and patients with other autoimmune diseases or infection. The presence and titer of FR and ACPAs have a diagnostic and prognostic role, as these autoantibodies are associated with a more severe disease phenotype, as well as increased extra-articular involvement of organs and tissues [47].

ACPAs are postulated to affect bone remodeling independently, with a prominent role concerning bone mass loss in the pre-clinical phase of RA, meaning that patients with the presence of these antibodies were without synovial inflammation [48]. The relation between in vivo increased osteoclastogenesis and systemic bone loss ACPAs was first demonstrated in 2012, following the comparison of the injection of purified human ACPAs, from the serum of patients with RA, into lymphocyte-deficient Recombination activating gene 1 (Rag1) –/– mice with control IgG injections. Hence ACPAs can directly bind to citrullinated proteins, such as vimentin, present on the surface of pre-osteoclasts, resulting in an increase in osteoclastogenesis and differentiation into mature osteoclasts [49].

Furthermore, the interaction between ACPA immune complexes and the receptor for IgG Fc γ receptor (FcγR) present on osteoclasts results in the activation of the spleen tyrosine kinase (SYK), leading to an increase in intracellular calcium concentration and subsequent activation of MAP kinases (MAPK) [50]. This signal transduction cascade is very similar to the one engaged by OSCAR activation, which could reasonably explain the direct pro-osteoclastogenic activity of these immune complexes. Finally, the interaction between these ACPA immune complexes and macrophage cells via FcγR results in the release of TNF-α [51], which, as extensively reviewed earlier, strongly contributes to the enhancement of bone resorption and serves as a bridge between autoimmunity, inflammation, and bone loss. Noteworthy ACPAs have been shown to have a significant clinical impact as an independent factor in systemic bone mass loss in various cohorts of early arthritis patients, comparing bone mineral density values between ACPA-positive and ACPA-negative individuals [52,53].

Recently Anti-carbamylated protein (CarP) antibodies have been identified in the serum of patients with RA. These antibodies target epitopes that have undergone post-transcriptional modification of carbamylation, a chemical reaction between lysine residues and cyanate forming homocitrulline [54]. Nevertheless, the current lack of standardized methods for their detection limits the execution of clinical studies to assess their diagnostic and prognostic role in this subset of patients.

These autoantibodies can exert in vitro increased resorptive activity in osteoclast cultures in the presence of CarP-IgG immune complexes at a permissive concentration of RANKL [55], consistent with a direct impact of these autoantibodies on bone remodeling, similar to what has been observed with ACPA. However, additional studies are warranted to explore this hypothesis further.

### 3.4. Bone Erosions in Rheumatoid Arthritis

Bone erosions represent central features of RA and are the primary manifestations occurring within the first 8 weeks from disease onset in as much as 10% of patients, with more than 50% of patients developing erosive RA within the first year from disease onset [56,57]. Bone erosions are typically recognized via plain radiographs as interruptions in the cortical bone with loss of adjacent trabecular bone in peculiar peri-articular locations where the synovium enters into direct contact with the bone, known as “bare areas” [58]. Similar to systemic bone loss, joint structural abnormalities observed in patients with RA result from an imbalance between upregulated mechanisms of bone resorption and downregulated mechanisms of bone formation.

The areas of periarticular bone loss are enriched in osteoclasts both in RA patients and animal models of arthritis [59,60,61] while showing a paucity in active osteoblasts [62]. Mechanisms of abnormal osteoclast activation in RA are multiple, closely mirroring RA pathogenic mechanisms [63], and involve interactions between autoantibodies, pro-inflammatory cytokines, immune cells, and synovium resident cells with the local bone microenvironment. Interestingly, not all the events that lead to bone resorption take place inside the joint cavity—within the synovial membrane. In fact, an additional stimulus to focal bone loss appears to come from the bone marrow niche surrounding the erosions, where a higher degree of marrow aggregates characterized by an increased number of osteoclasts adherent to the subchondral bone have been observed [64]. Before dissecting the role of the extracellular factors influencing osteoclasts’ abnormal activity, it is noteworthy that RA osteoclasts’ precursors per se demonstrate characteristics of enhanced cellular differentiation and activation [65]. In particular, both circulating pre-osteoclasts, synovial pre-osteoclasts, and peri-resorption sites’ mature osteoclasts in RA show an abundance of OSCAR expression [45,66], somehow suggesting that RA bone damage may be partly due to the OSCAR-mediated differentiation of imprinted osteoclasts’ precursors and excessive activation of pathologic mature osteoclasts [67]. In addition, the triggering receptor expressed on myeloid cells 2 (TREM2) and DAP12 were also found to be upregulated in inflammatory arthritis-derived osteoclasts ([46,68], explaining how priming of these cells is also partly mediated by an upregulation of the calcium-mediated auto-amplification of nuclear factor of activated T cells 1 (NFATc1) [69].

It is noteworthy that many of the mechanisms of action underlying ACPAs and pro-inflammatory cytokines in the pathogenesis of erosive damage are also involved in systemic bone loss, as reviewed in the previous section.

Approximately ten years ago, the first evidence of ACPAs binding to osteoclasts and the demonstration of bone loss induced by RA patients’ ACPAs transfer to mice was published [48]. It was demonstrated that the osteoclast binding ability was limited to IgG ACPAs, with polyclonal IgGs unable to exert the same osteoclastogenic capabilities. It was suggested that osteoclasts and osteoclasts’ precursors might represent preferential targets for ACPAs due to the physiologic display of PADs and citrullinated proteins by all osteoclast lineage cells [70], thus explaining the features of ACPAs-induced osteoclastogenesis before disease onset, even in absence of inflammation [49]. Recently, erosion was also described in ACPA-positive patients without clinical joint inflammation, but with tenosynovitis and osteitis on RMI imaging supporting its role as a biomarker of early RA [49].

Osteoclasts’ differentiation and activation promoted by autoantibodies occur through direct and indirect interactions. Mainly operating through the activation of the FcγR on osteoclasts’ lineage cells with the subsequent differentiation of osteoclasts’ precursors [71], another Fc-independent mechanism of direct ACPA-induced osteoclasts activation is mediated by ACPA-originating Fab fragments, which induce osteoclastogenesis presumably through their binding to citrullinated molecules expressed on the pre-osteoclasts surface [48].

RF does not seem to exert any direct effect on osteoclasts. However, the pentameric structure of its most common IgM isotype allows it to enhance the formation and stability of IgG-ACPAs-containing immune complexes, thus increasing the autoantibodies’ osteoclastogenic properties [72,73]. Coherently, a synergistic role for both autoantibodies was observed in a study in which a higher burden of erosive changes in patients with RA was associated with the concomitant presence of RF and ACPAs compared to ACPAs alone [74]. Another key factor influencing IgG ACPAs’ ability to induce osteoclast activation is Fc glycosylation. In particular, the lower levels of sialic acid residues (sialylation) on the terminal Fc glycan observed on RA ACPAs increase their affinity to FcγR and thus their erosive properties [75].

The indirect mechanisms of ACPA-induced osteoclastogenesis partially overlap with another mechanism of bone resorption, through the promotion of TNF-α by FcγR-expressing macrophages [75]. Moreover, macrophages are activated by ACPAs via binding to the citrullinated glucose-related protein 78 (cit-GRP78) [76]. Thus, the indirect ACPAs’ contribution to bone loss occurs through the release of inflammatory cytokines.

As already reported above, TNF-α is a key cytokine in RA pathogenesis and bone loss [77,78]. TNF-α exerts diverse effects on osteoclastogenesis. First, the cytokine increases the production of RANKL by osteocytes [31] and mesenchymal cells [58], triggering osteoclast activation. Furthermore, TNF-α and RANKL act in concert in the bone destruction process inducing downstream activating pathways such as NF-κB, activator protein (AP)-1 and NFATc-1, signaling mechanisms involved in osteoclastogenesis [79]. Lastly, another mechanism of TNF-induced bone loss is the increase in circulating osteoclast precursors through bone marrow hematopoietic stem cells’ fate commitment via the up-regulation of colony-stimulating factor-1 (c-CSF) expression [80]. Not only does TNF-α induce osteoclast proliferation and activation, but it also disrupts osteoblasts pathways through the degradation of the pro-osteoclastogenic factor Runx2 [81] and the induction of Wnt inhibitors DKK-1 and SOST by synovium-resident cells [37,82].

Macrophages, T cells and synovial fibroblasts are the major producers of synovial IL-6 [83]. Like TNF-α, IL-6 contributes to osteoclastogenesis, stimulating RANKL production by osteoblasts through signal transducer and activator of transcription 3 (STAT3) activation [84], although IL-6’s osteoclastogenic properties are still debated [85]. IL-6 shares another feature with TNF-α, which is the ability to suppress osteoblast-mediated bone formation. Both cytokines promote the expression of the metallopeptidase ADAMTS4, which in turn cleaves the osteoblast-enhancing semaphorin Sema3A in favor of the inhibitor semaphorin Sema4D [86].

Even though it was initially postulated as a lack in osteoclasts’ IL-1 signaling [87], there is a potential for IL-1 to replace RANKL during the late stages of osteoclast differentiation [88], while simultaneously reducing osteoprotegerin (OPG) production by osteoblasts [89]. The influence of IL-1 on bone loss mechanisms is paralleled by the observation of a reduction in radiographic scores with the use of IL-1β inhibitor Anakinra [90].

Last but not least, IL-17 plays a decisive role in RA synovitis and bone loss [91]. Whether a direct contribution of IL-17 on bone cells is exerted or not is still to be elucidated. The presence of RANKL and M-CSF was able to induce an IL-17-mediated osteoclast activation [92]. However, the same effect was not replicated by other observations [93]. Overall, IL-17 seems to play an indirect effect on osteoclastogenesis by increasing the production of pro-inflammatory cytokines by stromal and immune cells [94], by activating innate immune cells, osteoblasts, and synovial fibroblasts to express RANKL [95]. IL-17 is produced by a specific subset of CD4+ T helper cells, called Th17. CD4+ T cells infiltrate the RA synovium where they display multiple effector functions [96]. IL-17 expression is only one of the osteoclastogenic mechanisms offered by Th17 cells. These T helper cells are major actors in antibodies’ Fc desialylation [97], promoting immune complexes-mediated osteoclastogenesis. Th17 also reportedly increases pre-osteoclast recruitment through the production of chemokines by mesenchymal stromal cells (MSCs) located in the bone marrow [98]. Under normal conditions, immunological balance is established with the aid of FOXP3-expressing T regulatory (Treg) cells [99]. Other than suppressing inflammation, high amounts of IL-10 and Cytotoxic T-Lymphocyte Antigen 4 (CTLA-4) expressed by Treg act as inhibiting factors on osteoclasts and bone destruction [100]. In detail, CTLA4-expressing Treg cells bind to CD80/86 presented on osteoclast precursors’ surfaces, leading to intracellular activation of indoleamine-2,3-dioxygenase followed by the apoptosis of osteoclast precursor cells [101]. On the other hand, secreted IL-10 induces the upregulation of OPG and downregulates the production of RANKL [102]. Although committed, Treg cells show plastic properties and when exposed to an inflammatory environment, they lose their FOXP3 expression and transform into arthritogenic Th17 cells (called exFOXP3Th17 cells) [103]. These cells copiously produce cytokines and chemokines that can induce more profound osteoclastogenesis than naive CD4+T cells-derived TH17 cells [103,104]. Both Th17 and exFOXP3Th17 cells play a conspicuous role in RA bone loss by interacting with one of the major contributors to developing erosions, which are synovial fibroblasts.

Synovial fibroblasts (also renamed as type B fibroblast-like synoviocytes, FLS) are stromal tissue-resident MSCs that, under physiological conditions, cohabit the synovial lining layer with type A macrophages-like synoviocytes. In the lining layer, FLS and macrophage-like synoviocytes serve as a barrier function. Moreover, FLS more loosely populate the sub-lining layer among the less densely packed tissue matrix [105]. Synovial fibroblasts preserve the joint cavity with lubricating molecules and plasma-derived nutrients, as well as playing a landscaping function with the constantly fine-tuned regulation of the extracellular matrix (ECM) composition through the production of ECM components (such as collagens, fibronectin, vitronectin, and proteoglycans), ECM-degrading matrix metalloproteinases (MMPs), and their inhibitors [106]. Moreover, FLS act as immune sentinels, in that they display the features of innate immune cells such as Toll-like receptors (TLRs) 2, 3, and 4 through which FLS activate the classical NF-κB and AP-1 pathways to generate chemokines and MMPs [107]. Lastly, FLS serve as bridges between the innate and adaptive immune responses thanks to the expression of the molecule cluster differentiation 40 (CD40)—an important molecule expressed by antigen presenting cells (APCs) for their activation—through which they induce the production of pro-inflammatory cytokines and autoantibodies by activated T-cells and CD40-expressing B lymphocytes, respectively [108].

The synovium in RA patients undergoes profound hyperplastic modifications, with the thickening of the lining layer growing from the two-to-three-cells deep physiologic layer to an excessively proliferated layer of ten to twenty cells (both FLS and macrophages-like synoviocytes) in depth [105]. Moreover, at the synovium border, the thickened tissue transforms into a mass of “pannus” rich in FLS and osteoclasts that invades the adjacent articular cartilage and subchondral bone [109].

Such aggressive phenotypic modifications seem to be closely related to immune cell recruitment as well as to FLS phenotypic changes. FLS shows a state of overproliferation coupled with a pro-inflammatory, invasive phenotype that is maintained even throughout several cellular generations and in the absence of activating triggers as a persistently imprinted make-up [110].

Different subsets of RA-related FLS distinguished by the differential expression of surface markers have been identified and their aggressive phenotype is seemingly obtained through interactions in the inflamed synovial microenvironment [111]. Such FLS populations differentially localize in the lining and/or sub-lining and exert several pathological functions, although two main subsets exist: the sub-lining located inflammatory CD34-FAPα + THY1 + FLS, and the lining-resident tissue-destructive CD34-FAPα + THY1 − FLS [28]. Such major subpopulations account for the two principal mechanisms through which FLS mediate bone loss: a direct, RANKL- and MMP-mediated bone loss, and an inflammatory cytokines-driven bone injury.

The aggressive front of the synovial pannus is mainly composed of macrophages and fibroblasts that secrete tissue-degrading enzymes causing cartilage and bone damage [112]. Pathological, synovial lining-FLS shows a discrete pattern of surface protein expression. Under the influence of inflammatory cytokines these cells express high levels of membrane-bound podoplanin (PDPN) and CD55 that mediate features of migration, tissue aggression and invasiveness [111]. Moreover, the molecular analysis of the aggressive milieu in which lining-resident FLS operate demonstrates a niche in which pannus-resident FLS have protected against apoptosis thanks to the upregulation of the transcription factor p53 and the downregulation of the tumor suppressor PTEN [75]. Aggressive synovial fibroblasts produce a variety of MMPs to degrade ECM and articular cartilage tissue under the promotion of three main inducers: pro-inflammatory cytokines (among which IL-1 is probably the most potent inducer), growth factors like (FGF) and platelet-derived growth factor (PDGF), and matrix molecules such as collagen and fibronectin [113]. There is good evidence that transcription factors and promoter activators overlapping with immune functions (like various MAPK families such as ERK, JNK, and p38, as well as AP-1, NF-kB activators and STAT and ETS transcriptional factors) regulate the increased MMPs production by FLS [110,114].

In addition, synovial fibroblasts proved to be the main source of articular, soluble RANKL when activated by interaction with inflammatory cytokines [103]. In collagen-induced arthritis (CIA) models, mice lacking RANKL expression in FLS, but not those lacking RANKL expression in T- or B-cells, demonstrated protection against bone erosions [115]. These data indicate the FLS are the main producers of RANKL in inflammatory arthritis and support the concept of “tissue-destructive synovial fibroblasts” [116]. Another determinant contribution of FLS in bone damage is through the inhibition of osteoblasts’ bone-forming properties. It has been demonstrated that activated synovial fibroblasts produce DKK-1 from the earliest stages of disease [117], thus cushioning osteoblasts functions and enhancing the destructive profile of FLS. Moreover, it was observed that, in turn, DKK-1 exacerbated the inflammation and tissue-degrading enzyme production by FLS [118]. Other than DKK-1, FLS inhibit the Wnt pathway and osteoblast bone-forming properties by secreting a high amount of sclerostin [37], even though data regarding the inhibition or deletion of sclerostin in TNF-α-dependent models of arthritis (but not in TNF-α-independent arthritis models) induced acceleration of the synovial pannus formation and subsequent local inflammation and bone injuries, suggesting a more sophisticated role of sclerostin in the TNF-α signaling pathways [119].

As orchestrating homeostatic cells, physiological and pathological FLS show intricate interplay with various cellular and humoral environments. In particular, FLS are particularly sensitive to inflammatory cytokines. Macrophages- and TH17-derived TNF-α, IL-17, and IL-22 mediate the proliferation of FLS and promote the production of pro-inflammatory cytokines (such as IL-6, TNF-α, and IL-1) and chemokines by these cells [120]. The inflammatory milieu of the synovium promotes the activation of inflammatory pathways by FLS by upregulating the expression of cadherin-11 on fibroblasts’ membranes, which in turn creates abundant homotypic interactions between cadherin-11 molecules, thus activating the NF-kB pathway to increase the production of pro-inflammatory cytokines, in particular IL-6 [121]. Another mechanism of FLS inflammatory phenotype activation is mediated by the interaction of FLS-expressed CD40 with CD40L molecules on activated T cells. CD40-CD40L binding enhances the expression of IL-6 as well as adhesion molecules like intracellular adhesion molecule 1 (ICAM1) and vascular adhesion molecule 1 (VCAM1) [122]. The secretion of pro-inflammatory cytokines and adhesion molecules helps maintain an inflammatory microenvironment through a vicious circle amplified by FLS. IL-6 produced by FLS is both a key promoter of Th17 differentiation and an important molecule in the conversion of FOXP3+ T cells in exFOXP3Th17 cells, while in turn IL-17 further stimulates IL-6 production by FLS in a positive feedback loop renamed “IL-6 amplifier” [123]. Other than VCAM1 and ICAM1 enhancing the interactions between T cells and synovial fibroblasts [124], FLS participate in the recruitment of inflammatory infiltrates into the joint through the production of high levels of inflammatory chemokines [125]. Stimulated fibroblasts produce both neutrophil-attracting chemokines like chemokine (C-X-C motif) ligand (CXCL8), CXCL5, and CXCL1 [126], as well as CX3CL1 (fractalkine) recruiting CX3CR1-expressing T cells, CXCL10 recruiting CXCR3+ Th1 cells, and CCL20 recruiting Th17 cells to inflammatory synovium [127,128]. Furthermore, FLS promote the survival of B cells through the secretion of VCAM1 and CXCL12 [129], as well as through the B cells’ differentiation and production of the survival factors APRIL and BAFF induced by FLS-TLR3 ligands [130], thus enhancing the pathogenic roles of autoantibodies.

Notably, a particular macrophage subset expressing CX3CR1hiLy6CintF4/80+I-A/I-E+ termed arthritis-associated osteoclastogenic macrophages (AtoMs) have been identified as possible pathogenic osteoclast precursors in RA [131]. As such, FLS would further participate in synovial pathology and bone loss mechanisms via their enhanced production of CX3CL1, allowing for the migration of pathogenic osteoclast precursors to inflamed joints.

Lastly, a recently discovered mechanism of monocyte commitment towards the osteoclast lineage involves the participation of phagocytic, innate-immune cells neutrophils and their aberrant expression of neutrophils extracellular traps (NETs) [132]. Furthermore, in addition to the established role of NETs in perpetuating the exposure to post-translationally modified autoantigens and PADs in the synovial milieu [133], they promote tissue injury by releasing cartilage-degrading elastase [134] and by potentiating IL-17-mediated inflammation [135].

### 3.5. Phospholipase C Gamma and Rheumatoid Arthritis

Phospholipase C is a Ca++-dependent phosphodiesterase that acts as the primary effector in the inositol signaling pathway, as it catalyzes the synthesis of diacylglycerol (DAG) and inositol 1,4,5-triphosphate (IP3), two important second messengers, from phosphatidylinositol 4,5-bisphosphate (PIP2) [136]. It mediates a wide range of biological effects, including cell proliferation and migration, as well as angiogenesis, playing a significant role in oncogenesis, in addition to serving as a critical regulator of T cell receptor responses [137]. There are 13 different isoforms of this enzyme, further grouped into six defined families: PLC-δ, β, ε, γ, η, and ζ [138]. PLC gamma seems to be engaged in the osteoclastic differentiation process at multiple levels, both by mediating downstream signaling in the pivotal RANK-RANKL pathway crucial for osteoclastogenesis and by regulating adhesion receptor signaling during osteoclast development [139]. Nevertheless, the possible role of PLC gamma in pathological conditions characterized by an alteration in basal bone remodeling, such as in rheumatoid arthritis, remains to be elucidated, as prompted by both in vitro and in vivo data indicating that PLC plays a minor role in estrogen deficiency-induced bone resorption compared to the physiological bone resorption process [140].

### 3.6. Osteoporosis in Rheumatoid Arthritis

Osteoporosis is the most common skeletal disorder characterized by progressive decreases in bone mass and microarchitecture impairment, resulting in an increased risk of fractures [141].

Several studies have established a significantly increased risk of osteoporosis in patients affected by RA compared to healthy age–sex-matched controls. Furthermore, a meta-analysis of 25 cohort studies and almost 250,000 RA patients showed a 1.6-fold greater risk of fragility fractures [142,143].

The association between osteoporosis and RA is multifactorial and does not solely reflect the use of glucocorticoids, immobility, disability, and the consequent increased risk of falls, but also the specific role of rheumatoid arthritis as a systemic autoimmune inflammatory process that can lead to extra-articular involvement, including bone tissue [144]. Pro-inflammatory cytokines’ overexpression along with immune cells’ hyperactivation and the presence of specific autoantibodies account for this detrimental interaction between bone remodeling and rheumatoid arthritis through several pathogenic mechanisms that can enhance osteoclastogenesis, osteoclast lifespan and activity, and conversely can impair bone apposition.

## 4. Bone Involvement in Spondyloarthritis

SpA represent a group of interconnected chronic inflammatory conditions characterized by both peripheral joints and axial involvement. As in other rheumatic diseases, bone resorption markers are elevated, leading to bone loss and an elevated risk of vertebral fractures. On the other hand, SpA is also associated with new bone formation in the peripheral and axial joints, in particular, the so-called syndesmophytes result in increased rigidity and ankylosing of the spine. SpA also have a negative effect on bone: in ankylosing spondylitis (AS) elevated bone resorption leads to increased bone loss and, as stated, an increased vertebral fracture risk.

### 4.1. HLA-B27

HLA (human leukocyte antigen) class I molecules (A, B, and C), are expressed on the cellular membranes of nearly all nucleated cells. These molecules present endogenous antigens to cytotoxic T cells, facilitating the development of loss of tolerance to self-antigens [145]. It is widely recognized that SpA exhibits a genetic predisposition linked to the HLA-B27 allele [146]. This allele encompasses over 100 subtypes, which exhibit varying distribution patterns across diverse populations [147]. The prevalence and incidence of AS are positively correlated with the prevalence of the HLA-B27 allele in the population [148]. The highest prevalence of HLA-B27 is found in patients with ankylosing spondylitis (AS) and non-radiographic axial SpA (nr-axSpA) (75–90%) while patients with inflammatory bowel disease-related SpA (SpA-IBD) show the lowest prevalence of HLA-B27 (10–40%) [149].

Several research groups have explored the role of HLA-B27 in bone remodeling, both in normal physiological conditions and in the presence of inflammatory states. For instance, in HLA-B27+ transgenic mice, the administration of exogenous TNF-α leads to a 2.5-fold increase in osteoclastogenesis compared to wild-type mice [150]. In cells expressing HLA-B27, exposure to TNF-α leads to an overstimulated production of both IL-1α and IFN-β, suggesting an intricate interplay between TNF-α and HLA-B27. However, despite the dual secretion of pro-osteoclastogenic (IL-1α) and anti-osteoclastogenic (IFN-β) molecules after TNF-α exposure, the former appears to dominate, fostering an overall pro-osteoclastogenic environment [150]. Additional studies have unveiled a heightened production of pro-inflammatory cytokines in HLA-B27+ mice, correlating with decreased bone mass and strength [151]. This result has primarily been attributed to increased bone resorption rather than decreased bone formation. Other studies have also indicated that HLA-B27+ mice exhibit a reduced OPG/RANKL ratio and a state of osteoporosis [152]. Collectively, the studies mentioned above converge to demonstrate that HLA-B27 organisms are predisposed to an imbalance in the OPG/RANKL-related mechanisms of bone remodeling, heightened osteoclastogenesis, and elevated bone resorption. Furthermore, in transgenic mice a lower bone mass, trabecular bone density, and mineral-to-matrix ratio have been demonstrated, but higher levels of c-terminal telopeptide (CTX), pyridinoline/divalent collagen cross-link ratio (a marker of mechanical competence), proteoglycans (which inhibit matrix mineralization), and alveolar bone loss [153]. Other in vitro data seem to report a direct positive influence of HLA-B27 on osteoclastogenesis [154].

Molecules inhibiting OPG activity result in lower bone mineral density (BMD): as written above for RA; recently autoantibodies against OPG have also been found in SpA, correlating to lower BMD and a history of fractures [27]. Moreover, it has been showcased that elevated soluble RANKL (sRANKL) levels and an increased soluble RANKL/OPG ratio among AS patients were associated with decreased BMD values [155]. Furthermore, these findings correlated with radiological signs of disease damage, higher clinimetrics scores, CRP and other pathogenetical cytokines, providing support to the presence of a direct link between joint and systemic inflammation, altered levels of important mediators of bone homeostasis and osteoclast activation, and overall BMD.

### 4.2. Cytokines

#### 4.2.1. IL-23 and IL-17

Pro-inflammatory cytokines like IL-23, implicated in SpA pathogenesis, also contribute to bone loss [156]. Mainly produced by Paneth cells in the gut and joint macrophages, IL-23 triggers IL-17 and IL-22 production by Th17 cells, both involved in systemic inflammation. IL-17 relates to bone loss, while IL-22 is associated with osteoproliferation [157]. IL-23 drives T-helper cells to become IL17-producing Th17 in osteoblast-free cultures, fostering osteoclastogenesis via a RANKL-independent pathway [158]. Additionally, IL-23 prompts osteoclastogenesis and enhances RANK expression in osteoclast precursors [159]. However, several studies suggest that such a catabolic action of IL-23 is possible exclusively through the mediation of IL-17 since IL-23 alone demonstrates an inhibitory effect on osteoclasts [160]. IL-17A is primarily generated by Th17 lymphocytes, which indeed have demonstrated osteoclastogenic activity [24]. IL-17A is released in response to TNF-β, IL1β, IL21, and IL23 stimulation, but suppressed by IFN-α [161]. Experimental studies indicate that exposure to very low or high IL-17A concentrations triggers increased RANK expression and JAK2-STAT3 signaling activation in circulating human MSCs (hMSCs), promoting their differentiation into mature osteoclasts. Interestingly, intermediate IL-17A levels yield the opposite effect on osteoclast precursors [162]. Synergistically with TNF-α, IL-17A increases the production of IL-1a, IL-1b, and IL-6, which in turn stimulate osteoclast activity [163]. Finally, IL-17A induces the secretion of RANKL by osteocytes, with a consequential increase in the RANKL/OPG ratio and TNF-α concentration [162]. Interestingly, in transgenic mice in which the expression of IL17A was increased, no alterations in either the number or activity of osteoblasts and osteoclasts, or in bone mass, were found. This suggests that IL17A may only have a detrimental effect on bone in an inflammatory context, thus acting in concert with other pro-inflammatory cytokines, while on its own it is unable to alter bone metabolism. Research using mouse models of psoriatic arthritis (PsA) has revealed a correlation between elevated circulating IL-17A and bone loss [164]. In these models, IL-17A stimulates the RANKL production of affected skin cells and hinders osteoblasts and osteocytes through the SOST/DKK-1 pathway [165]. Additional support for IL-17A’s role comes from anti-IL17 treatments like secukinumab and ixekizumab used in PsA management, which bring about bone stability, decreased bone erosion, and reduced bone damage progression [166].

#### 4.2.2. TNF-α

TNF-α and IL-17A can mutually enhance each other’s effects, resulting in diverse outcomes in distinct settings. For instance, within synovial joints, the coexistence of osteoclasts (OCs), driven by elevated RANKL levels prompted by these cytokines, leads to bone erosions [167] but OCs’ differentiation could also be induced in a RANKL-independent pathway [168].

TNF-α amplifies the presence of circulating CD14+/CD11b+ osteoclast precursors both in transgenic mice featuring the persistent activation of TNF-α and in mice receiving TNF parenteral administration [169]. This correlation is further substantiated by evidence showing that the administration of TNF-α inhibitors (TNFi) to patients leads to a reduction in the count of circulating progenitors [170].

#### 4.2.3. Other Cytokines

Few data have been published on the role of growth differentiation factor (GDF)-15, also known as macrophage inhibitory cytokine 1 (MIC-1) in SpA, which appears to be elevated in the synovial fluid but not in the sera of SpA patients [171]. This cytokine is a member of the TGF-β superfamily but its pro-inflammatory or anti-inflammatory activity is still debated. It rises after a TNF-α stimulus, secreted by osteocytes after hypoxic damage, with consensual osteoclast activation [172]. This cytokine seems to be higher in SpA sera, with respect to healthy controls, and it is positively correlated with CRP levels, with the presence of erosion in the sacroiliac joint (SIJ), and with a high degree of X-ray scores [173].

Finally, in early SpA IL-31 seems to be associated with low BMD and lower syndesmophytes development, evaluated as a modified Stoke Ankylosing Spondylitis Spinal Score (mSASS) < 1. IL-31 is an IL-6 superfamily member and a Th1-Th2 border cytokine. It binds its receptor (heterodimer of IL-31RA and Oncostatin M) activating JAK/STAT, PhosphatidylInositol 3-Kinase/AKT (PI3K/AKT) and MAPK pathways. IL-31 is produced by memory Th2 lymphocytes, macrophages and dendritic cells. IL-31 activity on bone remodeling is still debated: It induces metalloproteinases and osteoclastogenic cytokines, inducing higher differentiation of osteoclast progenitors in mature cells [174], and it is correlated with pro-inflammatory cytokines in early SpA. However, some studies have demonstrated its anti-inflammatory action in different mucous membranes [175,176]. Its serum levels correlate with those of TNF-α and IL-6, but not of IL-17. A study demonstrated a correlation between IL-31 and DKK-1 levels, resulting in an inhibition of bone formation and activation of bone resorption by the RANK-RANKL pathway [177].

As opposed to RA, skeletal abnormalities observed along the SpA spectrum appear multifaceted, with simultaneous evidence of bone loss and bone formation (Figure 2).

A fundamental pathway enhancing osteoblast differentiation and activity revolves around wingless (Wnt). Two important down-regulators of the WNT/beta-catenin pathway are DKK-1 and SOST, although data on their serum levels in patients with SpA are conflicting. Both of them are directly related to CRP and inflammation but only SOST is minimally reduced under TNFi treatment [178]. Increased levels of expression of DKK-1 are observed both in the synovium and serum of mice subjected to increased levels of TNF-α, RANKL, and other inflammatory cytokines [32]. This suggests that DKK-1 is over-expressed under conditions of inflammation. The same study identified DKK-1 as a molecule capable of inducing osteoclastogenesis and, on the other hand, of down-regulating osteophyte formation, depending on the microenvironment. Indeed in enthesis, in mice transgenic for the constitutive expression of TNF, the inhibition of DKK-1 resulted in osteophyte production, while the simultaneous expression of both molecules preserved bone from osteoproductive damage [32]. Moreover, these results are supported in the same context by the finding of increased local and serum levels of osteocalcin and OPG when mice were treated with anti-DKK-1 antibodies. This observation lends further support to the existence of a cross-talk between DKK-1 and the RANKL-OPG system, suggesting an inhibitory effect of DKK-1 on OPG. The resorptive effect of DKK-1, in fact, seems to be driven by its inhibition of OPG. In mice treated with anti-DKK-1, the concurrent suppression of OPG leads to the reactivation of osteoclastogenesis. Conversely, when OPG is administered subsequently, this reactivation is once again halted. Interestingly, it is worth noting that these processes do not appear to influence osteophyte formation [32].

Other important molecules involved in osteoblastogenesis, as well as in endochondral ossification, are the so-called bone morphogenetic proteins (BMPs), which belong to the TGF family. These proteins, as well as the BMPs/DKK-1 ratio, were higher in the serum of patients affected by AS than in healthy controls, and positively correlated with disease activity, structural damage, and disease duration, suggesting that BMPs have a role in the formation of pathological new bone. Intriguingly, although in other studies DKK-1 levels were found to be lower in AS patients compared to healthy controls, a positive correlation between DKK-1 levels and the risk of vertebral fractures in AS patients was observed [179]. This underscores the complex interplay of these molecules in AS and suggests that the role of DKK-1 in bone metabolism is not straightforward and may involve intricate interactions with other factors. Because lower levels of DKK-1 and SOST were found in AS patients, the authors suggested that the main cause of osteoporosis in AS should be searched for in other alterations such as immobilization, systemic inflammation, and biomechanical factors.

While in RA the onset of a state of systemic osteoporosis seems to follow the “so-called” juxta-articular osteopenia, in SpA this has not been demonstrated [180]. An assay of the levels of OPG expressed by resident cell populations in the synovial joint membrane [181] confirmed such suggestion, demonstrating that OPG expression by endothelial cells and lining macrophages was absent in patients with active RA, contrary to what is observed in patients with SpA and even inactive RA. At the same time, the study demonstrated an increased expression of RANKL in both RA and SpA patients.

### 4.3. Erosions in Spondyloarthritis

Early research characterized SpA as a non-erosive inflammatory condition. However, subsequent research has changed this perspective, revealing that erosions can indeed occur even in SpA. It is worth noting that the data showing comparatively less local bone damage in SpA might be explained by various factors such as the direct joint damage mediated by autoantibodies in RA.

A clear elucidation of how the microenvironment driven by T helper cell polarizations can either activate or inhibit osteoclastogenesis was provided by Sato and colleagues in a previously mentioned in vitro study [24]. Co-cultures of osteoclast precursors and osteoblasts were established, introducing Th1 and Th2 cells to examine their effects on osteoclastogenesis, measured by tartrate-resistant acid phosphatase (TRAP) levels. Interestingly, the presence of Th1 or Th2 cells alone did not trigger osteoclastogenesis. Conversely, the introduction of Th17 cells, particularly after exposure to IL-23, stimulated osteoclastogenesis. This effect of Th17 cells is linked to IL-17, and the significance of this cytokine in osteoclast activation was further confirmed by the fact that mice lacking the IL-17 gene did not exhibit increased osteoclastogenesis [24]. Moreover, IL-23 itself has been demonstrated to induce osteoclastogenesis [182] and this agrees with the results of clinical trials in which the use of IL-23 inhibitors (IL-23i) or IL-17 inhibitors (IL-17i) antibodies prevent radiographic progression in PsA patients [182,183].

A clear illustration of the complex interplay between different pathways of innate immunity in the context of inflammatory arthritis emerged from a study in which adenoviral vectors containing genes for both IL-17 and TNF-α were introduced into the knee of a CIA mice model. The outcome demonstrated a more pronounced inflammation and increased bone destruction compared to the effects of each cytokine alone. This combined effect was also marked by a synergistic up-regulation of the NF-kB pathway, elevated MMP production, and more extensive cartilage damage. These findings were substantiated by their subsequent investigation involving the injection of IL-17R and TNF-binding protein (TNF-BP) to act as decoy receptors for these inflammatory cytokines. This intervention led to a substantial reduction in joint inflammation, further supporting the pivotal role of IL-17 and TNF-α in the inflammatory process. Importantly, the study revealed that only the concurrent presence of both cytokines resulted in irreversible bone and cartilage erosions, in contrast to TNF-α alone [184].

Psoriatic arthritis (PsA) is the more prevalent disease in the SpA family showing a tendency to periarticular bone erosions. The erosive disease seen during PsA is associated with higher levels of RANKL at the synovial lining layer, both absolute and relative to those of OPG, and with higher levels of circulating CD14+/CD11b+ pre-osteoclasts [170]. Some authors found in the serum of PsA patients, with and without erosive disease, higher levels of DKK-1 than in healthy controls [185]. In the same study, higher levels of M-CSF were found in the subgroup of PsA patients showing erosive disease. The number of circulating pre-OC and the levels of serum M-CSF and sRANKL were associated with radiographic damage scores, including erosions and osteolysis. A more recent paper confirmed the finding of higher serum DKK-1 in PsA patients and demonstrated that it represented a risk factor for bone erosions, with a significant OR of 4.4 [186]. Notably, studies are conflicting because some groups confirmed the higher levels of DKK-1 in PsA but others described even lower levels with respect to healthy controls [187].

Other cytokines involved in PsA pathogenesis and related bone disease are IL-20 and IL-24. These pro-inflammatory molecules seem to exert their effect on bone through the induction of MCP1, which is a recruiting cytokine for an osteoclast precursor [188]. The IL-20 family of cytokines is very large, including, other than IL-20 and IL-24, IL-19 (which exerts principally anti-inflammatory effects), IL-26, and IL-22. The latter is produced by Th22 and Th17 under IL-1b, IL-6, and IL-23 stimulation [188]. In particular, IL-22 showed damaging properties, being associated with joint destruction causing erosions in arthritic animal models [189]. Nevertheless, a definite role for IL-22 as a biomarker or therapeutic target in axSpA has yet not been identified [190], and its effects on bone do not appear to be solely confined to erosive properties.

### 4.4. New Bone Formation in Spondyloarthritis

New bone formation is a key process in the pathophysiology of SpA. It could affect both the axial joints (i.e., the SIJ and the spine) and peripheral sites, as well as periarticular tissues such as the enthesis. The molecular mechanisms of new bone formation have been largely studied in axSpA, while fewer data are available on PsA. To the best of our current knowledge, the most credited hypothesis is that the new bone formation process represents the final step in the continuum of SpA bone lesions. Indeed, bone overproduction generally occurs where bone marrow edema (inflammation) and, subsequently, fat metaplasia take place [191]. Nevertheless, two main controversial points persist. First, nearly one third of the patients affected by axSpA will never develop structural changes during the disease’s course [192]. Second, in some patients, new bone formation happens even when the disease is well controlled with optimal treatment and it could be partially explained if new bone formation was intended as a finalistic effort of the spine to increase its stability [193]. On the other hand, histological analysis of the spine and the SIJ of patients with AS detects inflammatory infiltrates in the subchondral bone marrow even where inflammatory lesions have not been seen via imaging techniques, suggesting that sometimes MRI might miss mild inflammatory changes [194,195]. Therefore, despite the arrival of biological DMARDs (bDMARDs) having dramatically improved the quality of life and the physical performance of patients affected by SpA, the relationship between new bone formation and chronic inflammation remains not completely understood. Therefore, while some studies suggest the existence of a coupling between inflammation—and disease activity—and new bone formation in patients with AS [196,197,198], others demonstrate that ossification progresses independently from the suppression of inflammation involving adjacent but distinct sites [199,200]. Moreover, most findings from clinical trials show less bone formation in patients with axSpA treated with bDMARDs whose disease is well controlled by the therapy, suggesting that new bone formation in the context of a clinically controlled disease could be explained by a subclinical, persistent, non-detectable joint inflammation [201,202].

### 4.5. The Enthesis Milieu

The enthesis is the site of the attachment of tendons, ligaments, fasciae, and joint capsules to the bone [203]. During the last couple of decades, several studies have contributed to evolving the concept of the enthesis from being considered merely as an anatomical site to a biologically complex entity including the tendon, the entheseal fibrocartilage, and the subchondral bone marrow [204]. Enthesitis, which is considered a hallmark of PsA and, in general, of all the SpA, is the result of the combination of the inflammatory process and a mechanical load. The relevance of enthesis in SpA and the key role of IL-17 and TNF-α were also confirmed in a study, which demonstrated a population of tissue-resident memory CD4 + and CD8 + T cells with an immunomodulatory phenotype but which were still able to produce IL-17 and TNF at the level of enthesis [205]. Although bone formation could be considered the final step of the inflammation and a mechanism of repair for bone loss, an interesting study demonstrated that new bone formation in the Achilles tendon of patients with PsA occurs at distinct sites from erosions [206]. The mechanical load takes part actively in the pathogenesis of enthesitis, as reported in a study involving transgenic mice prone to develop inflammatory features similar to SpA. Indeed, new bone formation was strongly associated with mechanical load and correlated with inflammation and interestingly the unloading of the enthesis significantly reduced the inflammation compared to weight-bearing mice [207]. Overall, the process of bone spurs’ formation at entheseal sites resembles a response-to-injury process, mimicking the healing from a fracture through a sequence of endochondral bone formation, where a cartilage scaffold is then replaced with bone tissue [208].

The new bone formation at entheseal sites has been further deepened by a Chinese study in which the ossifying spinal enthesis of 10 patients with AS and 10 controls were analyzed, finding a higher expression of BMP-2 and an enhanced osteogenic differentiation of hMSCs in the enthesis of AS patients compared to controls [209]. In particular, an imbalance between BMP-2 and its antagonist Nogging has been demonstrated in patients with ankylosing spondylitis [210].

Osteogenic precursor cells (OPCs), the MSCs responsible for new bone formation in SpA, could be resident or recruited, but a yet unsolved question is how recruited OPCs reach the enthesis and few data regarding this have been published. To this purpose, CXCL12 has been found to be overexpressed in spinal ligaments sampled from patients with AS compared to healthy controls and OPCs were attracted to sites where CXCL12 was overexpressed [211].

### 4.6. New Bone Formation Pathways

#### 4.6.1. Wnt/ß-Catenin

In both axial and peripheral SpA, the role of the pathways of Wnt-ßcatenin—and its regulator molecules—in the process of new bone formation has been extensively studied, most often with conflicting results.

During the last 15 years, several studies tried to analyze the role of DKK-1 in SpA pathogenesis. Higher levels of DKK-1 in the serum of patients with AS have been reported compared to RA patients and healthy controls and the addition of the serum of AS patients to lymphocytes cell culture promoted Wnt activity [212]. Despite this seeming paradoxical, a dysfunctional activity of DKK-1 could partially explain the results. However, a recent meta-analysis including 40 studies on the measurement of serum DKK-1 levels did not find any differences between AS patients and healthy controls and higher serum levels of DKK-1 did not increase the risk of developing AS [213]. Further, as demonstrated in other studies including cohorts of axSpA patients such as the DESIR cohort, DKK-1 levels were not related to structural damage nor with levels of DKK-1 single nucleotide polymorphisms (SNPs), suggesting that DKK-1 production is not genetically determined in patients affected by axSpA [214,215]. Interestingly, DKK-1 levels seem to correlate with the entity of bone marrow edema and seem to decrease in patients receiving TNFi, partially explaining the possible decrease of new bone formation under treatment with TNFi [216].

DKK-1 has also been studied in patients with peripheral SpA such as psoriatic arthritis—although less extensively—with controversial results even in these cases. An Italian study demonstrated lower serum levels of DKK-1 in patients with PsA compared to healthy controls and RA patients for the first time [187]. This evidence differs from other previous and subsequent studies in which—similarly to SpA—higher levels of DKK-1 in PsA patients were reported [185,217]. Furthermore, in an observational case–control study including 50 patients with PsA and 50 healthy controls, DKK-1 serum levels were significantly higher in PsA patients and positively correlated with both inflammation, erosions, and calcification at entheseal sites, evaluated by ultrasound using the Madrid Sonographic Enthesitis Index (MASEI) [218]. However, conclusive data on DKK-1 levels in patients with SpA are still lacking [178].

Data on SOST in patients with SpA are controversial as well. Interestingly, SOST seems to be overexpressed in HLA-B27-positive people, independently from a diagnosis of SpA [219], as well as in males and elderly SpA patients [214]. However, several studies found discordant results, finding SOST serum levels not significantly different between patients with AS and healthy controls [178], or even lower in patients with early axSpA in the DESIR cohort compared to healthy controls [214]. In the absence of specific serological markers useful for detecting subclinical inflammation in people with SpA under treatment, a study on 30 patients with AS treated with TNFi and 30 healthy subjects found that SOST serum levels were significantly lower in AS patients compared to controls and even lower in patients with elevated CRP, suggesting that lower levels of serum SOST could reflect the presence of underlying inflammation [220]. In addition, in a work conducted on serum and tissue samples from zygapophyseal joints of patients with AS, RA, OA, and healthy controls, a lower expression of SOST in both the serum and chondrocytes of patients with AS compared with OA and healthy controls was found. Furthermore, SOST levels were reported to be inversely correlated with syndesmophyte formation [221]. Moreover, an Italian study reported lower levels of SOST and higher anti-SOST-IgG serum levels in patients with axial SpA-IBD compared to patients with only IBD and they negatively correlated with the duration of musculoskeletal symptoms [222].

As well as for AS, the role of SOST in the pathogenesis of new bone formation in patients with PsA is not univocally demonstrated. Most of the studies reported higher levels of serum SOST in PsA patients compared to healthy subjects, even if a correlation with disease activity or with structural damage had not been demonstrated [223,224]. In conclusion, although the reduced activity of DKK-1 and SOST in patients with SpA could be assumed to be a relevant mechanism of new bone formation, current evidence based on in vitro and on human biological samples are discordant and not conclusive.

Indian Hedgehog (IHH) is produced by chondrocytes and it is an important regulator of endochondral ossification [225], mechanistically similar to bone formation in SpA. However, the role of IHH in SpA has not been widely evaluated so far and only one study analyzed the serum levels of IHH of patients with AS, RA, and healthy subjects, and the effect of such serum on chondrocyte cell cultures, finding higher levels of IHH in patients with AS in comparison with AR patients, as well as in AS patients treated with TNFi compared to those not receiving therapy [226]. Furthermore, the molecule antagonist noggin inhibits the onset and progression of spontaneous arthritis and endochondral bone formation in a preventive and therapeutic way. Endogenous noggin seems to affect the progression of joint remodeling and slows the ossification process [227].

#### 4.6.2. Cytokines

Many cytokines participate in the pathogenesis of SpA and are targeted by bDMARDs. As discussed above, the relationship between inflammation and new bone formation is still debated, and how pro-inflammatory cytokines could promote a different bone phenotype in SpA and RA (Table 2) is only partially known.

TNFi has been the first class of bDMARDs approved for treating SpA. TNF is a pro-inflammatory cytokine which binds two receptors: TNFR1 and TNFR2. While TNFR1 delivers pro-inflammatory and catabolic stimuli, on the other hand, TNFR2 has opposing functions. Therefore, while the TNFR2-mediated signaling increases new bone formation via the inactivation of osteoclasts, TNFR1 promotes bone loss [228]. As mentioned above, treatment with TNFi seems to increase bone mass in patients with SpA but as TNF plays an important role in the inflammatory process in patients with AS, its effect on bone formation and ankylosis is still not understood, even though early initiation and long-term treatment with TNFi was demonstrated to slow radiographic progression [229,230].

Despite the evidence of the direct and indirect pro-osteoclastogenetic properties of IL-17 and its role in systemic and focal bone loss in RA and SpA (discussed above), surprisingly, several studies report IL-17 as a promoter of osteoblastogenesis, leading to higher bone formation, probably via a JAK2/STAT3 pathway [231]. Regardless, the overall in vivo effect of cytokines and other biological regulators is hard to analyze because many other factors take part in the microenvironment of the joint and the enthesis of patients with SpA. A very interesting study conducted on bone marrow samples of healthy subjects demonstrated that IL-17A acts as an osteogenic factor, stimulating the osteogenic differentiation of human bone marrow-derived mesenchymal stem cells (MSCs) and blocking RANKL, assisted by TNF [167]. Overall, the biological net effect of IL-17 in patients with SpA could seem contradictory. However, a possible explanation for these conflicting results could reside in the stage of differentiation of bone-forming cells exposed to IL-17 and the timing of their exposure to the cytokine, as IL-17 acts as a stimulator of MSCs’ osteoclastogenic differentiation [167,232], while it exerts anti-differentiating effects on osteoblast-committed cells [233].

**Table 2 biology-12-01320-t002:** Effect of several endogenous proinflammatory substances on bone metabolism in RA and SpA.

Molecule	Disease	Effect on Bone Metabolism
TNF-α	RA and SpA	Binding to TNFR1: Promotion of osteoclastogenesis increasing RANKL expression in osteocytes [29,222] Promotion of osteoclastogenesis stimulating IL-1 secretion [29] Binding to TNFR2 (in SpA): In enthesis milieu, reduction of bone loss inhibiting osteoclasts’ activity [222]
IL-1	RA	Promotion of osteoclastogenesis acting on bone marrow-derived macrophages [29]
IL-6	RA	Promotion of osteoclastogenesis increasing RANKL expression in osteoblasts [85]
IL-17	✓ RA ➢ SpA	✓ Promotion of osteoclastogenesis stimulating the production of other pro-inflammatory cytokines [24,94] and increasing RANK expression in osteoclasts precursors and RANKL secretion by osteocytes [157] ➢ Promotion of the differentiation of osteoblasts precursors, synergistically with TNF-α [162]
IL-23	SpA	Stimulation of polarization into Th17. [182] Promotion of osteoclastogenesis increasing RANK expression in osteoclast precursors [27]
IL-22	SpA	Promotion of osteogenetic differentiation and migration of osteoblasts precursors synergistically with IFN-γ and TNF [234,235]

DKK-1: Dikkopf-1; FcγR, fc gamma receptor; IL: interleukin; RA: rheumatoid arthritis; RANK: receptor activator of NF-κB; RANKL: receptor activator of NF-κB ligand; SOST: sclerostin; SpA: spondyloarthritis; TNF-α: tumor necrosis factor-α; TNFR: tumor necrosis factor-α receptor; IFN-γ: Interferon gamma.

Translating these findings to clinical practice, the advent of IL-17i and IL-12/23i created interest in the potential inhibition of both inflammation and new bone formation, allowing us to delay radiographic progression. The relevance of IL-17 as an osteogenic stimulus has been confirmed first in vitro, demonstrating that IL-17i could reduce both the inflammation and the ossification process [234], then in clinical trials of secukinumab and ixekizumab, two IL-17i, which have been approved for the treatment of axSpA and PsA and have been demonstrated to be effective in achievement of clinical remission and in reducing new bone formation in the spine [235]. However, the inhibition of IL-23 signaling in patients with axSpA did not achieve the expected results. Indeed, the ustekinumab (IL-12/23i) and risankizumab (IL-23i) trials on ankylosing spondylitis have failed and it is probably explained by the very early intervention of IL-23 in the pathogenesis of axSpA and by the evidence that IL-17 could be produced independently of IL-23 regulation [236,237,238].

IL-22 is another important cytokine in SpA. It is a pleiotropic cytokine with an important function in the regulation of adaptive immunity [239]. Serum levels of IL-22 were increased in patients with axSpA—particularly in patients with longstanding disease—compared to healthy subjects in a recent study. However, IL-22 serum levels were not associated with disease activity nor with radiographic progression [190]. These results do not differ from what has yet been demonstrated. Indeed, the concept is that IL-23-mediated synthesis of IL-22 is believed to contribute to new bone formation [240]. Interestingly, in an in vitro study, IL-22 drove the migration and the osteogenic differentiation of hMSCs only when IFN-γ and TNF were added to the culture, demonstrating once again how biological processes are regulated by highly complex systems [241].

Apart from the classic regulators of bone metabolism, the potential role of many other molecules has been studied. The most important are represented by adipokines—such as leptin—and serotonin. Leptin is an adipokine produced by adipocytes that seems to promote osteoblast and inhibit osteoclast activity in vitro [242]. However, studies on leptin levels in patients with SpA are once again not conclusive. Data from the ENRADAS trial reported a protective role of leptin against radiographic progression [243]. Interestingly, considering the usually higher levels of leptin in women, it has been suggested that leptin could partially explain the lower rate of new bone formation in women with axSpA [243].

Serotonin is a polyhedral substance produced by both the nerve cells in the central nervous system and the enterochromaffin cells of the pancreas. Among its functions, gastrointestinal serotonin acts as an inhibitor of osteoblasts’ proliferation in vitro and, consequently, of the bone formation process [244]. A study published a few years ago demonstrated lower levels of serotonin in patients with AS compared to RA and even lower levels in patients with AS under treatment with TNFi [245]. Such findings gave way to the acknowledgement of the so-called gut–bone axis and the potential role of serotonin in the regulation of bone metabolism in patients with SpA has been hypothesized.

### 4.7. Osteoporosis in Spondyloarthritis

Various studies indicate that the prevalence of osteoporosis in SpA ranges from 12% to 34% with a four-fold increase in the risk of vertebral fractures compared to healthy individuals [246]. Specifically, factors contributing to the occurrence of fractures include male gender, longer duration of the disease, concurrent inflammatory bowel disease, elevated disease activity scores, reduced femur BMD, and the presence of syndesmophytes [247]. The role of skin involvement as a risk factor for the development of osteoporosis is still debated. A recent study conducted on a large cohort of SpA patients found no disparity in the prevalence of osteoporosis based on phenotypes (axial or peripheral) in the presence of skin psoriasis [248]. Among spondyloarthritis related to inflammatory bowel disease (IBD), it can be suggested that a heightened risk of osteoporosis exists, most likely due to malabsorption issues stemming from these gastrointestinal disorders.

## 5. Conclusions

To conclude, while the underlying mechanisms of bone erosions have been extensively elucidated, giving some convincing mechanistic evidence especially about RA pathogenetic processes, the bone involvement in SpA is still far from being fully elucidated.

Given the extensive literature concerning the impact of pharmacological therapies on bone in these conditions [249,250], we have opted to omit this aspect in our review.

No bone markers have been identified to help in the clinical management of nor provide prognostic certainties on the radiographic progression in both RA and SpA.

Further studies need to clarify the interplay between bone tissue and rheumatic chronic arthropathies.

## Figures and Tables

**Figure 1 biology-12-01320-f001:**
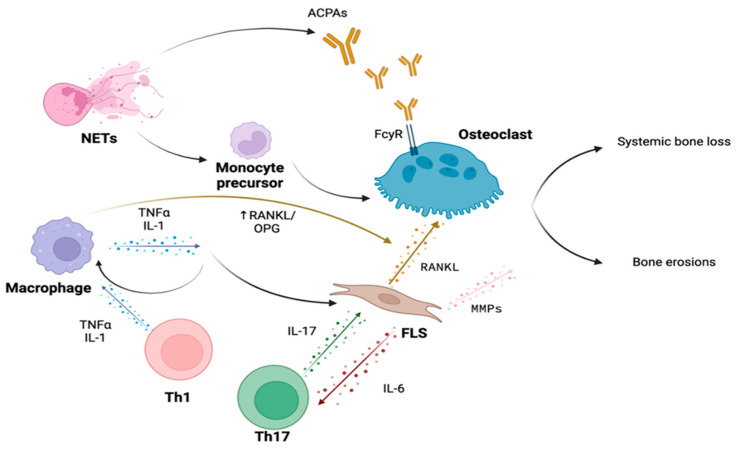
Cellular and molecular interplay involved in the pathogenesis of bone damage in rheumatoid arthritis. ACPAs, anti-citrullinated protein antibodies; RANKL, receptor activator of nuclear factor kappa-Β ligand; OPG, osteoprotegerin; TNFα, tumor necrosis factor-alpha; FLS, fibroblast-like synoviocytes; NETs, neutrophil extracellular traps; MMPs, matrix metalloproteinases; FcγR, fc gamma receptor; IL-1, interleukin-1; IL-6, interleukin-6; IL-17, interleukin 17.

**Figure 2 biology-12-01320-f002:**
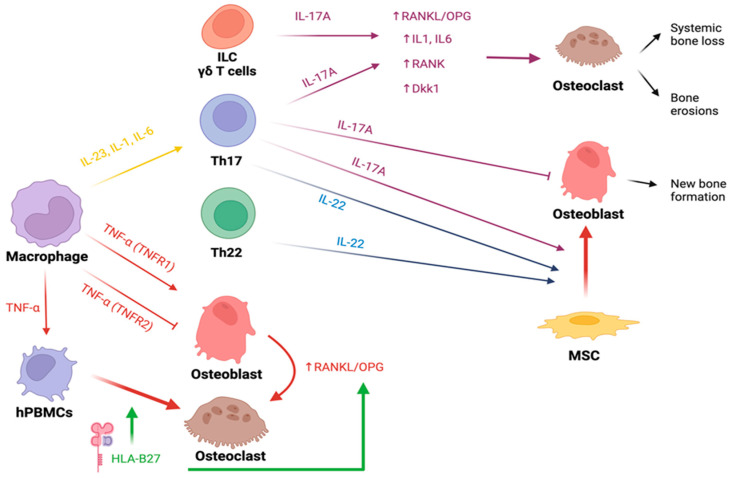
Interaction between the immune system, osteoclasts, and osteoblasts in the pathogenesis of bone involvement in spondyloarthritis. MSC, mesenchymal stromal cells; ILC, innate lymphoid cells; hPBMCs, human peripheral blood mononuclear cells; HLA-B27, human leukocyte antigen-B27; RANKL, receptor activator of nuclear factor kappa-Β ligand; OPG, osteoprotegerin; DKK1, Dickkopf-related protein 1; TNFα, tumor necrosis factor-alpha; TNFR1-2, tumor necrosis factor receptor 1-2; IL-1, interleukin-1; IL-6, interleukin-6; IL-17A, interleukin-17A; IL-22, interleukin-22; IL-23, interleukin-23.

**Table 1 biology-12-01320-t001:** Effect of immune cells on bone metabolism in RA and SpA.

Cells	Effect on Bone Metabolism
DC	Differentiate into osteoclast-like cells stimulated by RANKL and M-CSF [23]
Th1 cells	Reduce osteoclastogenesis inhibiting RANK expression [23]
Th2 cells	Reduce osteoclastogenesis inhibiting RANK expression [23]
Th17 cells	Dual effect on bone metabolism (bone loss/bone formation) depending on the microenvironment, via IL-17 production
Th23 cells	Polarization into Th17 via IL23 production [27]
FLS	Promote bone loss via both RANKL-dependent and RANKL-independent pathways [28]

DC: dendritic cells; FLS: fibroblast-like synoviocytes; IL: interleukin; M-CSF: macrophage colony stimulating factors; RA: rheumatoid arthritis; RANK: receptor activator of NF-κB; RANKL: receptor activator of NF-κB ligand; SpA: spondyloarthritis; Th: T helper.

## Data Availability

Not applicable.

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
