# Peer review of "Bone Involvement in Rheumatoid Arthritis and Spondyloartritis: An Updated Review"

_biology, 2023, doi:10.3390/biology12101320_

Round 1
Reviewer 1 Report
The authors identified a preponderance of the evolutions of inflammatory bone diseases that are associated with osteolysis, from the avascular (synovial-intra-articular) area to parts with rich bone vascularization. The association between this model of evolution and the possibility of inflammation to produce an increase in extracellular osmotic pressure with a tendency to compress adjacent areas (especially if they are septated or compartmentalized) suggests a decrease in the diffusion of mediators involved in extracellular signaling.
The authors identified common signaling pathways between the processes of the immune system and the processes of the osteosynthesis-osteolysis system. These ways are common due to evolutionary processes that have constantly tried to increase efficiency (energy, signaling, differentiation, transmission of genetic information, etc.). Although there are signaling molecules that overlap between the two systems, under physiological conditions, there are no erroneous interpretations of the signaling pathways. This leads to the hypothesis that there are multiple signaling pathways at this level, through changes in the weights of the intracellular signaling pathways (which are usually much more complex than the extracellular communication pathways).
The two major elements associated with the elements identified by the authors, (1) the decrease in the speed of extracellular signaling and (2) the common signaling pathways associated with differences in the complexity of intracellular and extracellular signaling, suggest that there is a time coding of extracellular signaling in addition to the existence of molecules signaling. This phenomenon is not unique in medicine, and is best known at the level of sexual hormones (where a pulsating level has opposite actions to a similar but uniform average level). An element that strengthens this assumption is the involvement of mechanoreception in bone remodeling processes, which is generated in pulses (mainly by locomotion).
Author Response
- We thank you for your thoughtful comments and valuable insights. We appreciate your time and expertise in reviewing our research.
Reviewer 2 Report
1. What are the Effects of Biological/Targeted Therapies on Bone Mineral Density in Inflammatory Arthritis?
2. What are the Bone phenotypes in rheumatology?
3. What are the Molecular Mechanisms of Joint Destruction and Pharmacological Treatments?
4. Role of DKK-1 (dickkopf-related protein 1) as a New Bone Formation Factor in Patients with Early Spondyloarthritis?
Author Response
- What are the Effects of Biological/Targeted Therapies on Bone Mineral Density in Inflammatory Arthritis?
We thank you for the suggestions. As requested by the Editor and the Guest Editors, the purpose of this review was to provide accurate and extensive evidence specifically regarding the pathogenetic mechanisms underlying erosive and osteoproliferative processes, that manifest differently in rheumatoid arthritis and spondyloarthritides. Consistently, we have omitted the aspect related to the impact of drugs used in these diseases, including both DMARDs and treatments for the bone involvement. This is because it falls outside the scope of our review. However, in accordance with your remark, we have added the bibliographic reference of a comprehensive review on this topic (Page 24, line 40).
- What are the Bone phenotypes in rheumatology?
Thank you for your suggestion. In the introduction, we have added a brief paragraph to provide a concise description of the various bone phenotypes involved in rheumatoid arthritis and spondyloarthritides (Page 2, line 11).
- What are the Molecular Mechanisms of Joint Destruction and Pharmacological Treatments?
We thank you for your comment. As reported above, the focus of our revision was to deal with the pathogenesis of bone damage in all its phenotypes, both in rheumatoid arthritis and spondyloarthritides. Beyond the involvement of the entheses that represent a key issue in the pathogenesis of the spondyloarthritides, we did not focus on the impact of the autoimmune process on other joint structures, as these aspects are not the primary subject of our review. The same applies to pharmacological therapies and their impact on joint damage.
- Role of DKK-1 (dickkopf-related protein 1) as a New Bone Formation Factor in Patients with Early Spondyloarthritis?
As reported in our paper, DKK-1 is an important regulator of bone metabolism and its impact on immune cells and pro-inflammatory cytokines has been largely discussed. However, following your suggestion, we added a citation on the correlation between Dkk1 and bone edema in early spondyloarthritis (Page 20, 42).
Reviewer 3 Report
The review is very interesting, but requires major revision:
- A table reporting the role of the different subset of immune cells on bone cells will lead a better undestanding of the paper;
- in the paragraph on RA a subparagraph on PLCgamma should be added;
- line 77: keep attention sclerostin is mainly secreted by osteocytes, not osteoclasts
- line 189: something else on romosozumab should be added;
- line 403: CD40 should be defined
- line 787: appear PsA, the author should check if it is correct. If yes, they should better explain the different subgroups included in the reported study;
- line 747: the paragraph number is incorrect and should be adjusted: possibly also evaluate to change the subtitle names in the description of RA and PsA;
- orthographic errors: line 155 a comma should be adjusted; line 394: a full stop is missed; round parenthesis should be eliminated at line 913, 935;
- references: ref 200 lacks; ref 145 should be completed; ref 155 has some strange characters; in general same journal name are reported in extenso others as abbreviations: should be adjusted
Author Response
The review is very interesting, but requires major revision:
We thank you for your revision that substantially improved our manuscript.
- A table reporting the role of the different subset of immune cells on bone cells will lead a better undestanding of the paper.
Following your suggestion. two tables have been added: one table recaps the role immune cells on bone metabolism (Page 4, line 9), the second one focuses on the role of various cytokines (Page 22, line 4).
In the paragraph on RA a subparagraph on PLCgamma should be added.
Following your remark, in the paragraph on RA a subparagraph on PLCgamma has been added (Page 13, line 16).
- line 77: keep attention sclerostin is mainly secreted by osteocytes, not osteoclasts.
We apologize for the error and we have modified the paper accordingly (Page 2, line 51).
.
- line 189: something else on romosozumab should be added;
We thank you for the precise input. Our review solely focused on the pathogenetic aspects related to bone damage, both focal and diffuse, in the context of RA and SpA. Romosozumab was the only drug mentioned based on scientific speculation regarding its potential exacerbation of rheumatoid arthritis. We wanted to clarify, as stated in Page 5, line 43, that in both the registration study and a recent meta-analysis, this side effect has not been demonstrated. As further evidence (page 5, line 44) we reported a clinical study where no inferiority in romosozumab's efficacy in GIOP treatment in RA compared to denosumab was demonstrated.
- line 403: CD40 should be defined.
Thank you for your suggestion. CD40 definition has been added (Page 11, line 12 ).
- line 787: appear PsA, the author should check if it is correct. If yes, they should better explain the different subgroups included in the reported study.
We edited the sentence reporting that enthesitis is the hallmark of SpA, including PsA, explaining why both SpA and PsA are cited in the paragraph.
- line 747: the paragraph number is incorrect and should be adjusted: possibly also evaluate to change the subtitle names in the description of RA and PsA;
We apologise for the mistake. All the paragraph numbers have been corrected.
- orthographic errors: line 155 a comma should be adjusted; line 394: a full stop is missed; round parenthesis should be eliminated at line 913, 935;
We apologise for the mistakes. The paper has been entirely reviewed and all orthographic errors have been fixed.
- references: ref 200 lacks; ref 145 should be completed; ref 155 has some strange characters; in general same journal name are reported in extenso others as abbreviations: should be adjusted.
Thank you for your remarks. We edited the present version of our paper after careful revision of all the references.
Round 2
Reviewer 3 Report
The authors addressed this reviewer's concerns.